# SLC7A11 Expression Is Up-Regulated in HPV- and Tobacco-Associated Lung Cancer

**DOI:** 10.3390/ijms252413248

**Published:** 2024-12-10

**Authors:** Julio C. Osorio, Cristian Andrade-Madrigal, Tarik Gheit, Alejandro H. Corvalán, Francisco Aguayo

**Affiliations:** 1Laboratorio de Oncovirología, Departamento de Ciencias Biomédicas, Facultad de Medicina, Universidad de Tarapacá, Arica 1000000, Chile; jcosoriop@academicos.uta.cl (J.C.O.); cristian.andradem@gmail.com (C.A.-M.); 2Epigenomics and Mechanisms Branch, International Agency for Research on Cancer (IARC), 69007 Lyon, France; gheitt@iarc.who.int; 3Department of Hematology & Oncology and Advanced Center for Chronic Diseases (ACCDIS), Pontificia Universidad Católica de Chile, Santiago 8320000, Chile

**Keywords:** human papillomavirus, lung cancer, SLC7A11

## Abstract

This study investigated the presence of human papillomavirus (HPV) in lung cancer patients from Chile. We found that HPV is present in 4% of lung carcinomas and that the SLC7A11/xCT gene is frequently up-regulated in HPV-positive lung carcinomas. Additionally, SLC7A11 is positively regulated in tobacco-smoke-associated lung cancer. This study suggests that SLC7A11 overexpression is associated with both HPV infection and smoking, with this up-regulation linked to poor survival rates in lung cancer patients.

## 1. Introduction

Lung cancer is a heterogeneous group of malignancies and the leading cause of cancer-related deaths in men and women worldwide [1]. Tobacco smoking (TS) is the most significant risk factor for lung cancer development, though additional risk factors such as viral infections have been suggested [2]. High-risk human papillomaviruses (HR-HPVs) have been found in a high percentage of lung carcinomas worldwide, though their role in this malignancy has not yet been clarified [3]. HPV is an epitheliotropic virus belonging to the *Papillomaviridae* family, with more than 200 characterized genotypes. Twelve of these HPV types have been classified as high risk for cancer development, including cervical, anogenital, and oropharyngeal cancers [4]. Indeed, HR-HPV has been found in lung carcinomas at highly variable frequencies worldwide [2,5]. Importantly, HPV detection rates have revealed significant regional variability. In Europe, these rates have exhibited considerable fluctuations, ranging from exceptionally high levels in certain countries, such as Spain (56%), to negligible or nonexistent levels in others. In Asia, particularly in China and Taiwan, high detection rates surpassing 50% have been observed. Countries in the Americas have reported a wide range of HPV prevalence, with some studies indicating a substantial presence and others reporting no detectable levels [5]. HR-HPV E6 and E7 are the viral oncoproteins involved in the capacity of this virus to immortalize human cells, evade apoptosis, and modulate cell proliferation, ultimately contributing to the development and progression of HPV-associated malignancies [6]. The ability of these oncoproteins to promote cell alterations is based strictly on their ability to interact with cellular partners [7]. The presence of HR-HPV has been previously reported in lung carcinomas from Chile. HPV16 was identified as the most frequent genotype, found in an integrated physical status, albeit with a low viral load [8]. Although some studies have reported the presence of HPV in non-smoker patients with lung cancer [9], HPV is frequently detected in smoker patients with this malignancy [10]. In addition, tobacco smoke can modulate the behavior of viruses and their relationship with the host, as has previously been reported in cervical cancer [11]. Thus, the role of HPV in the context of tobacco-smoke-driven lung cancer is unknown. In this study, we used transcriptomic approaches to characterize changes in the expression profile of lung cancer associated with both tobacco smoke and HPV. We found that solute carrier family 7 member 11 (SLC7A11/xCT), an antiporter that mediates the uptake of extracellular cystine, is frequently enhanced at transcript levels in HPV-associated lung carcinomas. This up-regulation was also observed in smoker patients with lung cancer, as well as in lung cancer cells ectopically expressing HPV16 E6 and E7. Interestingly, HPV+/SLC7A11-positive cases exhibited significantly poorer survival rates when compared to HPV-/SLC7A11-negative cases. Thus, SLC7A11 up-regulation can be a potential biomarker of survival in HPV/tobacco-smoke-associated lung cancer.

## 2. Results

### 2.1. Clinicopathological Features of Lung Carcinomas from Chile

In this study, 204 lung carcinomas consisting of 105 lung adenocarcinomas (AdCs) and 99 lung squamous cell carcinomas (SQCs) from Chilean patients were analyzed. No differences in age range (*p* = 0.498) or smoking habits were found (*p* = 0.3339). Additionally, a significant difference was found in the differentiation grade among SQCs and AdCs (*p* = 0.01831). The clinicopathological features of specimens used in this study are shown in Table 1.

### 2.2. HPV16 Is Detected Associated with Tobacco Smoking in Lung Carcinomas from Chile

Lung carcinomas were analyzed for HPV presence by multiplex PCR/Luminex. We found that 8 out of the 204 (4%) lung carcinomas were HPV positive. All of them were identified as HPV16 positive. No statistically significant differences were found among age (*p* = 0.3187), differentiation (*p* = 0.6952) and histological type (*p* = 0.3936). Conversely, a statistically significant difference was found in smoking (*p* = 0.0352) when HPV-positive and -negative cases were compared. The clinicopathological features of the specimens used in this study are shown in Table 2.

### 2.3. In Silico Analysis Reveals SL7A11 Up-Regulation in Lung Carcinomas from Smoker Subjects

Supervised significance analysis of microarrays (SAM) showed that the SLC7A11 and CYP1B1 genes were frequently up-regulated in the bronchial epithelium of smokers (GSE 7895 library). When large and small airway epithelial cells were analyzed, we found that the C3, EGF, SERP1, SLC7A11 and SYPL1 genes were up-regulated (Figure 1A and 1B, respectively). Thus, we selected SLC7A11 for subsequent analysis in lung carcinomas.

### 2.4. SLC7A11 Is Up-Regulated in HPV16-Positive Lung Carcinomas from Chile

Since HPV16 was detected in only eight lung carcinomas from Chile, we randomly selected 24 cases from 196 HPV-negative cases showing information regarding smoking habits for subsequent SLC7A11 analysis and comparisons. Thus, we established a sub-cohort of 32 HPV-positive and HPV-negative lung carcinomas. In this sub-cohort, we found that SLC7A11 was more frequently detected in HPV-positive lung carcinomas when compared to HPV-negative cases (*p* = 0.0080) (Figure 2, Table 3).

### 2.5. SLC7A11 Is Up-Regulated in Smoker Patients with Lung Cancer from Chile

We considered the same sub-cohort of 32 lung carcinomas. Via RT-qPCR, we found that among these patients, eight showed SLC7A11 expression, while 24 were negative for SLC7A11 transcripts. In this sub-cohort, we found a significant association between smoking status and SLC7A11 gene expression. Indeed, SLC7A11 transcripts were more frequently detected in smokers than in non-smokers with lung carcinoma (*p* = 0.0498, Table 4).

### 2.6. SLC7A11 Transcripts Are Up-Regulated in Both HPV-Positive and Tobacco-Smoke-Associated Lung Carcinomas from the TCGA Database

To compare the SLC7A11 gene expression in HPV16-positive vs. non-viral lung SQCs, we collected information from The Cancer Genome Atlas (TCGA) using the Oncodate program (https://oncodb.org/, accessed on 15 September 2024). We observed a statistically significant increase in SLC7A11 gene expression in HPV-positive cases when compared to HPV-negative cases (*p* = 0.0131, Student T test) (Figure 3A). To determine the effect of tobacco smoking on SLC7A11 gene expression in lung cancer, we used the UCSC Xena program with information from the GDC TCGA Lung Cancer Database. We found that cigarette smoke significantly increased SLC7A11 gene expression (*p* = 0.0266, Figure 3B). By analyzing gene expression data from a public database, we found that HPV-positive lung SQCs significantly increased SLC7A11 expression when compared to HPV-negative laryngeal squamous cell carcinomas (LSCCs), cervical squamous cell carcinomas (CSCCs) and oropharyngeal squamous cell carcinomas (OPSCCs). Furthermore, our analysis revealed that this enhanced SLC7A11 expression was particularly pronounced in lung SQC patients with a history of heavy smoking (Figure 3C).

### 2.7. SLC7A11 Transcripts Are Up-Regulated in Lung Cancer Cells Ectopically Expressing HPV16 E6/E7 Oncoproteins

BEAS-2B bronchial cells were stably transfected with either the retroviral vector pLXSNHPV16E6/E7 or the corresponding empty vector. HPV16 E6/E7 transcripts were detected only in those cells transduced with the pLXSNE6/E7 vector (Figure 4A,B). Next, we evaluated the levels of SLC7A11 transcripts in these cells. We found that the HPV16 E6/E7 transcript levels increased 3-fold in transfected BEAS-2B (E6/E7) cells when compared to BEAS-2B cells transfected with an empty vector (Figure 4C). Furthermore, both BEAS-2B cells expressing HPV16 E6/E7 and BEAS-2B cells exposed to CSC showed a significant increase in SLC7A11 expression. However, when E6/E7-expressing cells were exposed to CSC, no additional increase in SLC7A11 was observed (Figure 4D).

### 2.8. Survival Is Increased in HPV-Negative/SLC7A11-Negative Lung Cancer Patients

The clinical records of the sub-cohort of 32 lung cancer patients showed that HPV positivity significantly reduced survival rates. Patients with high levels of SLC7A11 expression also displayed decreased survival. Conversely, HPV-negative/SLC7A11-negative patients showed a statistically significant increase in survival (*p* = 0.0253, Figure 5). Additionally, other comparations were made between HPV absence (−) SLC7A11 absence (−) and HPV absence (−) SLC7A11 presence (+) and HPV presence (+) SLC7A11 absence and HPV presence (+) SLC7A11 presence (+) (Figure 5B,C). All of the comparative analyses revealed the influence of SLC7A11 on the survival outcomes.

## 3. Discussion

Lung cancer is a heterogeneous disease etiologically associated with tobacco smoke consumption, though both arsenic and radon exposures are additional risk factors for the development of this malignancy [12]. Viral infections have also been suggested as factors potentially involved in lung cancer [13]. Indeed, HR-HPV has been detected in a subset of lung carcinomas worldwide, though its specific role remains unclear. In this study, we found HPV16 presence in 4% of lung carcinomas from Chile. This frequency is lower than that previously reported in the same country in 2007 [14]. Although both analyses were carried out in different patients at different times, methodological differences related to the sensitivity of PCR most likely accounted for such differences. HPV presence has been detected in lung carcinomas worldwide, with a prevalence of 10–20% in most studies [15]. The most common HPV types detected in lung carcinomas are HPV16 and HPV18, the HR-HPV types most frequently associated with cervical cancer [16]. However, additional HPV types, such as HPV6 and HPV11, have also been detected in lung cancer specimens [17]. Importantly, it is thought that HPV can be more strongly associated with well-differentiated histological types [14]; however, in this study, HPV16 presence was not associated with differentiation status. Additionally, the signaling pathways that occur in cervical carcinogenesis [18] have been identified, with the viral oncoproteins E6 and E7 playing an important role [19].

An interesting concern is whether a potential interaction between HPV and tobacco smoke may contribute to tobacco-driven lung carcinogenesis. We have previously identified this type of cooperation in lung and cervical cancer cells in vitro [11,20,21]. Considering that tobacco-positive and tobacco-negative lung carcinomas are different clinical entities [22,23], a recent study revealed significant differences in the genomic alterations between non-smokers and smokers with lung adenocarcinoma from Chile. Non-smokers had higher overall genomic alteration frequencies (58% vs. 45.7%, *p*-value < 0.01), particularly in genes like EGFR, KRAS, and PIK3CA. However, smokers exhibited a more complex genomic profile, with alterations in a wider range of genes. The most notable differences were observed in EGFR (higher in never-smokers), PIK3CA and ALK (higher in never-smokers), and KRAS and MAP2K1 (higher in smokers). Alterations in these genes primarily consisted of somatic mutations in EGFR and fusions in ALK, while PIK3CA, KRAS, and MAP2K1 were mainly associated with mutations [24].

Interestingly, in this study, HPV16 presence was significantly associated with tobacco smoking, leading us to characterize the expression profile of tobacco-associated lung cancer and lung cells using in silico approaches. Thus, we found that SLC7A11/xCT is frequently up-regulated in both smokers and HPV16-positive lung cancer patients. Indeed, the increased expression of SLC7A11 had been previously observed in lung AdCs from smoker subjects [25]. However, the SLC7A11 protein showed low expression levels in HPV-positive head and neck squamous cell carcinoma (HNSCC) when compared to HPV-negative cases [26]. SLC7A11 belongs to the solute transporter (SLC) family of proteins and is located at the 4q28-q32 locus. It encodes for a 501 amino acid protein that forms the light chain of the Xc-system [27]. By 4F2 heavy-chain binding, SLC7A11 forms a heterodimer that provides cystine for glutathione synthesis, promoting the obligatory exchange of extracellular cystine for glutamate [28]. SLC7A11 gene expression is regulated by physiological conditions such as hypoxia, inflammation, and the increased production of reactive oxygen species (ROS) [29]. Indeed, tumor cells require high amounts of glutathione for controlling the oxidative stress to which they are exposed [30].

Given that SLC7A11 emerged from in silico data, we analyzed a sub-cohort of 32 lung carcinomas from Chilean patients (24 HPV-negative and 8 HPV-positive) to assess SLC7A11 gene expression. Interestingly, we found that SCL7A11 gene expression was significantly up-regulated in smokers and in HPV-positive Chilean patients. In fact, additional studies reported that SLC7A11 was overexpressed in cancer, including lung AdC [31]. The connections between SLC7A11 and immune function have been explored as well, suggesting a complex interplay between this gene and the antitumor immune response [32]. For instance, certain immune cells such as CD8+ T cells might trigger ferroptosis in cancer cells by reducing the production of SLC7A11 [33]. The exact mechanisms underlying the increased SLC7A11 gene expression in HPV-negative or tobacco-associated lung carcinomas are unknown. However, there is evidence that SLC7A11 expression is increased by known oncogenic pathways, including the Mitogen-Activated Protein Kinase (MAPK)/RAS, Phosphatidylinositol 3-kinase (PI3K)/AKT, and the nuclear factor kappa light chain enhancer of activated B cells (NF-κB) pathways [34,35]. Of note, these pathways are commonly activated in lung cancer, which in turn can lead to increased SLC7A11 expression through diverse mechanisms, including transcriptional activation or increased mRNA stability [28,36]. Importantly, it has been proven that HR-HPV is involved in the activation of these signaling pathways through the expression of E6 and E7 oncoproteins in cancer cells [37,38]. E6 and E7 can exacerbate glutaminolysis in cervical cancer cells, although only E7 can increase SLC7A11 gene expression, thereby increasing the malignant phenotype [39]. In this study, we have shown that HPV16 E6/E7 expression correlated with enhanced levels of SLC7A11 transcripts in lung cells, although no additional increase was observed in the presence of cigarette smoke condensate. Furthermore, in vitro studies revealed that the up-regulation of SLC7A11 resulted in diminished PD-L1 expression and attenuated ferroptosis cell death in A549 cells [40].

Interestingly, cumulative evidence has shown that tobacco smoke or its components are also involved in the activation of the glutamine pathway as the primary energy source supporting cell proliferation [39,41,42,43]. Thus, it can be speculated that SLC7A11 overexpression in both HPV-positive and tobacco-smoke-associated lung carcinomas is mediated by the activation of the MAPK, PI3K, or NF-κB pathways. On the other hand, hypoxia induces SLC7A11 gene expression by promoting hypoxia-inducible factor 1 (HIF-1) transcription factor activation in breast cancer [44]. In fact, HIF-1 is a master regulator of cellular responses to hypoxia, and it can directly bind to the third intron of the SLC7A11 gene to activate its transcription after exposure to paclitaxel, a chemotherapy drug, in breast cancer cells [45]. Thus, increased SLC7A11 expression can influence redox balance and cellular responses to oxidative stress [46].

In this study, we found increased survival in HPV-negative and SLC7A11-negative patients with lung cancer. Paradoxically, it has been reported that HPV-positive lung carcinomas are less aggressive and more responsive to chemotherapy treatment than HPV-negative cases [47]. The same observation has been reported in HPV-positive oropharyngeal cancer treatment [48]. Conversely, patients with high SLC7A11 levels had worse outcomes [49]. SLC7A11 is indeed a promising anticancer target for drug development against Smoker+/HPV+/SLC7A11+ lung carcinomas. SLC7A11 plays a crucial role in maintaining intracellular glutathione levels, protecting cells from oxidative stress [50]. This is particularly important for cancer cells, which experience increased oxidative stress. Both smoking and HPV infection are known to contribute to lung cancer development [16] and they can potentially influence the expression and activity of SLC7A11, making it a relevant target for this specific patient population.

In fact, it was reported that KRAS-mutant non-small cell lung cancer (NSCLC) can increase its chemoresistance by overexpressing SLC7A11, leading in turn to poorer therapeutic outcomes [51]. Of note, evidence shows that the SLC7A11 protein is involved in ferroptosis, a type of programmed cell death, and that tumors with low SLC7A11 levels were enriched in pathways promoting survival [46]. In this regard, HPV-positive lung carcinomas are more sensitive to ferroptosis, a type of programmed cell death [52] triggered by the accumulation of lipid reactive oxygen species (ROS) in a Fe^2+^ dependent manner [53]. Indeed, SLC7A11 protects cells from ferroptosis by absorbing cystine, which is used to produce glutathione, an antioxidant involved in ROS depletion [46]. Alterations in SLC7A11 gene expression may also sensitize HPV-positive lung cancer to certain chemotherapy drugs, such as cisplatin [54]. Thus, SLC7A11 can protect cancer cells from cisplatin by absorbing cystine for cysteine production during lung cancer treatment [34].

High SLC7A11 expression has been associated with tumor metabolism, which is highly dependent on glutamine and glucose for nutrient supply. Tumor cells with high SLC7A11 expression are highly dependent on glucose for survival [55]. In conclusion, this study reports that the SLC7A11 antiporter protein is frequently up-regulated in lung cancer and is correlated with both the presence of HPV and tobacco smoke consumption. Additionally, the SLC7A11 gene expression levels correlate with HPV16 E6/E7 overexpression in lung cancer cells (Figure 6). More studies are warranted to dissect the role and mechanisms involved in SLC7A11 overexpression in HPV-positive and tobacco-smoke-associated lung carcinomas. We suggest a model in which HPVE6/E7 are involved in such an increase in SLC7A11, although additional experimental approaches in lung cells are warranted to confirm this possibility.

## 4. Materials and Methods

### 4.1. Clinical Specimens

We used 204 previously collected formalin-fixed paraffin-embedded (FFPE) lung carcinomas obtained between 2012 and 2016 from the Pathological Anatomy Service of the National Thorax Institute, Santiago, Chile. The histological analysis was carried out by an experienced histopathologist. The clinical data were obtained from the pathology reports and the patients’ clinical records. This study was approved by the Ethics Committee Board of the University of Chile (N° 027; 25 May 2022). A sub-cohort of 24 HPV-negative lung carcinomas were randomly chosen to evaluate SLC7A11 gene expression.

### 4.2. DNA Extraction

FFPE tissues were incubated with digestion buffer (10 mM Tris-HCl pH 7.4; 0.5 mg/mL proteinase K, and 0.4% Tween 20) for 8 h at 56 °C with stirring. Subsequently, the samples were incubated at 95 °C for 10 min, centrifuged for 2 min at 16,000 g and stored at 4 °C until use. The amplification of a fragment of the β-globin gene was used to determine the quality of the DNA. The primer sequence was as follows: PCO3: 5′-ACACAACTGTGTTCACTAGC-3′ and PCO4: 5′-CAACTTCATCCACGTTCACC-3′. The amplification program was as follows: denaturation at 95 °C for 5 min; 45 cycles with a cycling profile at 95 °C for 30 s, 52 °C for 30 s, and 72 °C for 30 s; and a final extension at 72 °C for 5 min.

### 4.3. Multiplex HPV Genotyping

Multiplex Type-Specific E7-Based PCR (E7-MPG) was performed using a Luminex bead-based platform [56,57] at the International Agency for Research on Cancer, in Lyon, France. The multiplex Luminex hybridization assay enabled the detection of 21 HPV types including HPV16. Oligonucleotide probes with an amino group at the 5′ end for HPV16 types were coupled to carboxylated beads using the carbodiimide procedure described by Schmitt et al., 2006 [58].

### 4.4. Cell Culture and Transfections

BEAS-2B (normal bronchial tissue, CRL-9609) and A549 (lung adenocarcinoma, CCL-185) cells were directly obtained from the American Type Culture Collection (ATCC) (Manassas, VA, USA) and maintained in Dulbecco-modified Eagle medium (DMEM), supplemented with 10% inactivated fetal bovine serum (FBS) (Hyclone), 0.1 μg/mL gentamicin (Invitrogen), 1 U/mL penicillin and 1 μg/mL streptomycin (Invitrogen), and incubated at 37 °C in a 5% CO_2_ atmosphere incubator. The cells were checked for mycoplasma infection using standardized protocols. For transfection, BEAS-2B and A549 cells were transfected with the retroviral vectors pLXSN and pLXSN containing the HPV16 E6/E7 ORFs according to previously published protocols [21]. Cells were maintained in culture medium without antibiotics for 12–18 h, after which transfected cells were selected via the addition of 0.2 µg/mL puromycin (Gibco, Carlsbad, CA, USA).

### 4.5. RNA Extraction, cDNA Synthesis, and RT-qPCR

RNA was isolated from cells using TRIzol^®^ reagent (Thermo Fisher Scientific, Inc., Waltham, MA, USA), according to the manufacturer’s protocol. Following chloroform purification and isopropanol precipitation, RNA was suspended in DEPC water and stored at −80 °C until use. The RNA was treated with RQ1 RNase-free DNase (Promega, Madison, WI, USA) at 37 °C for 120 min and was incubated with RQ1 DNase Stop Solution for 10 min. cDNA was prepared in a 20 ml reaction volume containing DNAse-treated RNA (1 µg), 1 unit of RNAse inhibitor (Promega, Madison, WI, USA), 0.4 µg/mL random primers (Promega, Madison, WI, USA), 2 mM dNTP (Promega, Madison, WI, USA) and 10 units of Moloney Murine Leukemia Virus reverse transcriptase (Promega, Madison, WI, USA). The reaction mixture was incubated for 1 h at 37 °C. RT-qPCR was carried out using the primers in Table 1. The β-actin mRNA levels were used for the normalization of RNA expression. The amplification conditions were 95 °C for 5 min, followed by 35 cycles of denaturation at 95 °C for 15 s and annealing at 62 °C for 15 s. The dissociation temperature range extended from 63 °C to 95 °C. To confirm the amplified products, 2.5% agarose gel electrophoresis was carried out.

### 4.6. Gene Expression Analysis in Databases of Lung Cancer Samples

The SLC7A11 expression profile was evaluated using the Gene Expression database of Normal and Tumor tissues 2 (GENT2) (http://gent2.appex.kr/gent2/, accessed on 15 September 2024). It was also evaluated in NSCLC (AdC, LCC, SQC) with GENT2. We used the Oncogenomic Database (OncoDB) to identify abnormal gene expression patterns in SLC7A11, as well as in HPV-16 infections correlated to the clinical features of lung cancer. To identify abnormal gene expression patterns in SLC7A11 as well as a history of tobacco smoking, we used data obtained from The Cancer Genome Atlas (TCGA) dataset and Xena Functional Genomics Explorer. Finally, we performed gene expression (SLC7A11) meta-analysis using the Lung Cancer Explorer (https://lce.biohpc.swmed.edu/lungcancer/, accessed on 15 September 2024) through a forest plot to summarize the normal standardized mean difference for the tumor vs. meta-analysis.

### 4.7. Microarray Expression Data

In total, the data of 104 HPV-positive/-negative samples and smoker/non-smoker samples were extracted from the Gene Expression Omnibus (GSE 7895 library). SAM and hierarchical clustering analysis (HCL) were performed with the Multiple Experiment Viewer (TMeV) software (https://webmev.tm4.org/about, accessed on 15 September 2024). The gene ontology analysis of significant up/down-regulated genes and pathways was performed with the FatiGo software (https://pubmed.ncbi.nlm.nih.gov/14990455/, accessed on 15 September 2024). Additionally, each sample from both the HPV and smoker libraries was classified according to tissue type. Only supervised analyses based on the HPV/smoker condition were performed using the HPV/smoker libraries. Significant matched genes between the HPV and smoker libraries were analyzed to identify simultaneously up-regulated genes.

## 5. Conclusions

Our findings suggest a complex interplay between HPV, tobacco smoking, and SLC7A11 gene expression in the development and progression of lung cancer. We have demonstrated a significantly higher prevalence of SLC7A11 overexpression in HPV16-positive lung carcinomas compared to HPV16-negative tumors, particularly in patients with a history of heavy smoking. Furthermore, our in vitro experiments (BEAS-2B) implicate HPV16 E6/E7 oncoproteins in the up-regulation of SLC7A11 transcripts. Our clinical data also indicate a poorer prognosis for lung cancer patients with both HPV infection and smoking habits. These results highlight the potential use of SLC7A11 as a prognostic biomarker and therapeutic target in a subset of lung cancer patients, particularly those with HPV and smoking-related disease. Future studies are warranted to elucidate the precise mechanisms underlying the interaction between HPV, tobacco smoke, and SLC7A11 in lung carcinogenesis. 

## Figures and Tables

**Figure 1 ijms-25-13248-f001:**
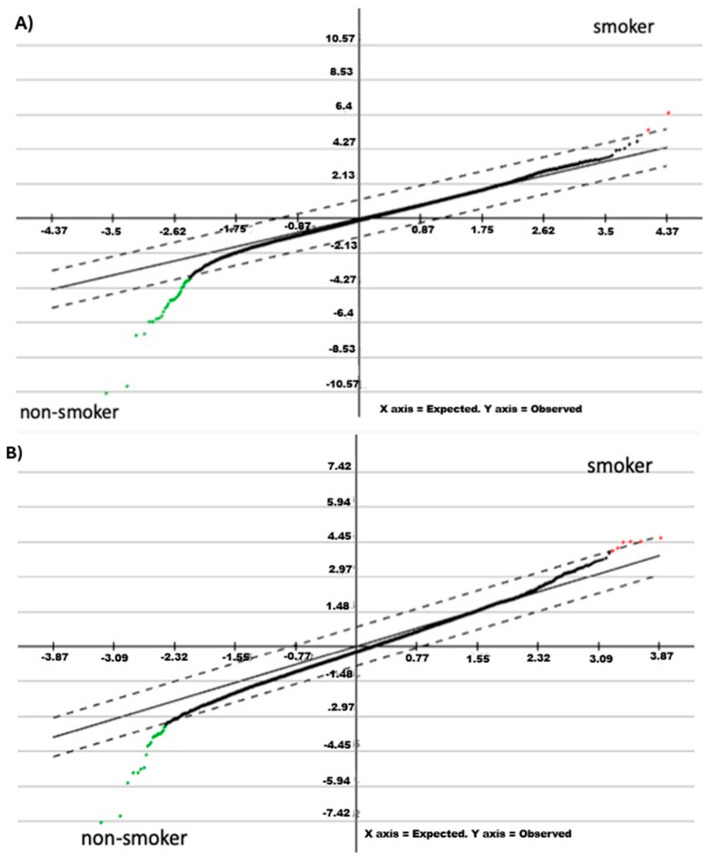
(**A**) Significance analysis of microarrays (SAM) showed that SLC7A11 and CYP1B1 are both up-regulated in the bronchial epithelium of smoker subjects (GSE 7895 library; 104 samples). (**B**) SAM showed that C3, EGF, SERP1, SLC7A11 and SYPL1 are up-regulated in large and small airway epithelial cells (GSE 10135 with 92 samples; and GSE 5060 with 124 samples). Overexpressed genes are illustrated in red; downregulated genes are illustrated in green.

**Figure 2 ijms-25-13248-f002:**
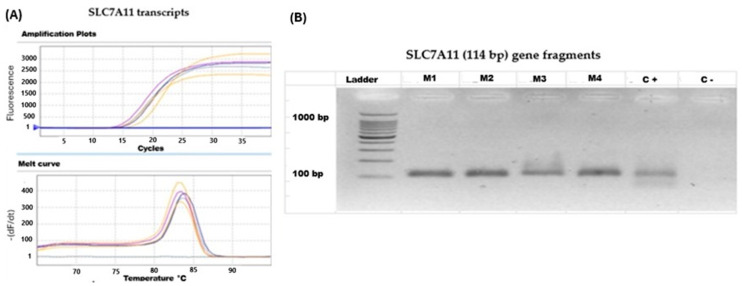
SLC7A11 gene expression in lung carcinomas from Chilean patients. (**A**) RT-qPCR for SLC7A11 transcripts in four lung carcinomas (M1-M4 clinical samples). (**B**) The 2.5% agarose gel electrophoresis of SLC7A11 (114 bp) gene fragments (M1-M4: clinical samples; C+: positive control; C-: negative control; ladder: 100 bp ladder).

**Figure 3 ijms-25-13248-f003:**
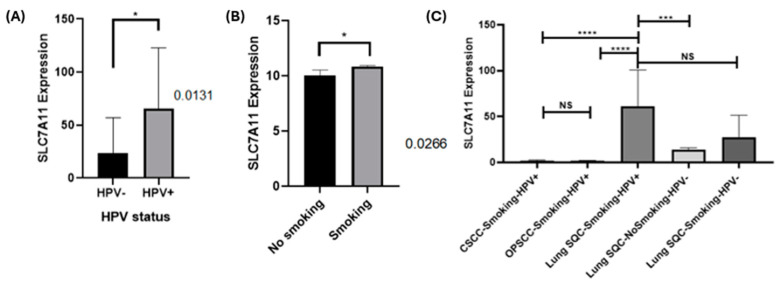
HPV16 E6/E7 expression correlates with the increased SLC7A11 expression levels in lung cancer. (**A**) Gene expression level comparison of SLC7A11 in HPV16 vs. non-viral lung SCCs, with the five-hundred and three Lung Cancer Database using the oncodb web source. (**B**) SLC7A11 expression and tobacco smoking history in lung carcinomas from the GDC TCGA Lung Cancer Database using UCSC Xena web source (right). In total, 290 lung carcinomas were selected. *: *p* ≤ 0.05. (**C**) SLC7A11 expression and tobacco smoking history (smokers categorized as severe) in lung SQC in relation to the presence of HPV. We used other types of cancer as controls (CSCC and OPSCC).***: *p* ≤ 0.001; ****: *p* ≤ 0.0001.

**Figure 4 ijms-25-13248-f004:**
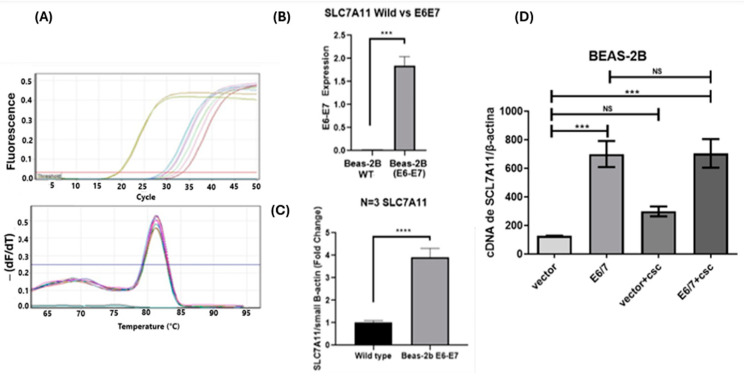
HPV16 E6/E7 expression correlates with increased SLC7A11 expression levels in lung cells. (**A**) HPV16 E6/E7 transcripts were evaluated by RT-qPCR in BEAS-2B wild-type cells and BEAS-2B cells stably expressing HPV16 E6/E7; melting analysis showing HPV16 E6/E7 transcripts; (**B**) HPV16 E6/E7 expression by RT-qPCR in wild-type BEAS-2B and BEAS-2B E6/E7 cells; (**C**) SLC7A11 expression by RT-qPCR in wild-type BEAS-2B and BEAS-2B E6/E7 cells normalized against beta-actin transcripts (Fold change). (**D**) RT-qPCR with primers for E6/E7 transcripts from BEAS-2B cells, only with a vehicle or with exposure to 10 μg/mL CSC and an equivalent DMSO concentration. ***: *p* ≤ 0.001; ****: *p* ≤0.0001.

**Figure 5 ijms-25-13248-f005:**
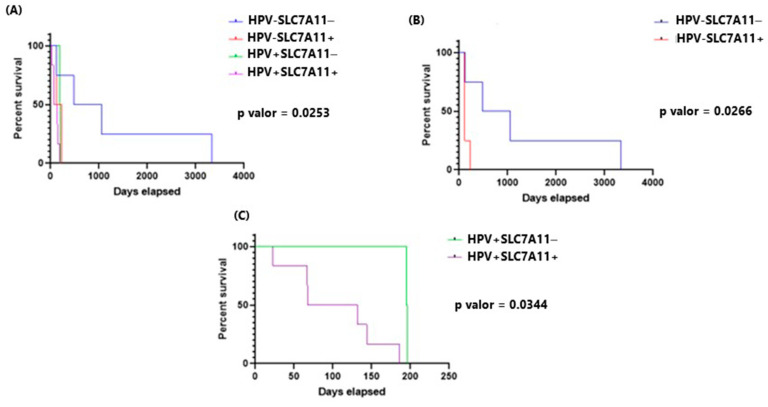
Kaplan–Meier survival of patients with lung cancer from Chile. (**A**) HPV absence (−) SLC7A11 absence (−), HPV absence (−) SLC7A11 presence (+), HPV presence (+) SLC7A11 absence, HPV presence (+) SLC7A11 presence (+). (**B**) Evaluation of only HPV absence (−) SLC7A11 absence (−) and HPV absence (−) SLC7A11 presence (+). (**C**) Only were evaluated HPV presence (+) SLC7A11 absence and HPV presence (+) SLC7A11 presence (+).

**Figure 6 ijms-25-13248-f006:**
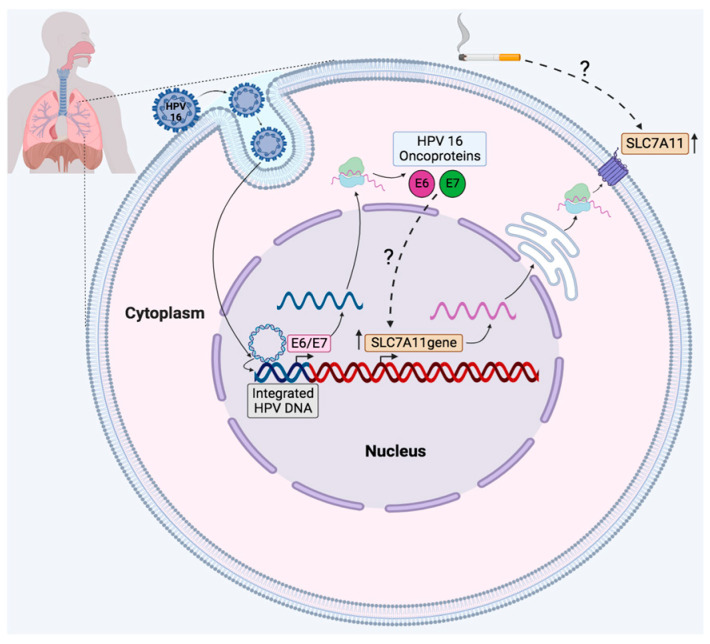
A hypothesized model of HPV16 E6/E7 promoted SLC7A11 overexpression in lung cancer. Both HPV and tobacco smoke increase the levels of SLC7A11 in lung epithelial cells. The overexpression of HPV16 E6/E7 oncoproteins and the activation of oncogenic signaling pathways are involved in SLC7A11 overexpression. (?): The mechanisms by which tobacco increases SLC7A11 levels are unknown. Constructed by BioRender.

**Table 1 ijms-25-13248-t001:** Clinicopathological features of lung SQCs and AdCs.

Histological Type	SQC	AdC	*p*-Value
	N (%)	N (%)	
**Total**	99 (46)	105 (54)	
**Age**			0.4980 ^&^
≤65 years	35 (35)	40 (38)	
>65 years	64 (65)	65 (62)	
**Differentiation**			0.0118
Poor	64 (63)	56 (53)	
Moderate	35 (37)	42 (40)	
Well	0 (0)	7 (7)	
**Smoking habit**			0.3339 ^#^
Smoking	31 (31)	29 (28)	
Non-smoking	50 (25)	63 (36)	

^&^ Fisher’s Exact test; ^#^ Chi Square test.

**Table 2 ijms-25-13248-t002:** HPV16 presence in 204 lung cancer carcinomas from Chilean patients.

		HPV16 Presence		
	(−) Cases (%)	(+) Cases (%)	Total	*p*-Value
**TOTAL**	196 (96)	8 (4)	204	
**Age**				
≤65 years	70 (36)	4 (50)	74	
>65 years	126 (64)	4 (50)	130	
**Smoking habit**				
Smokers	56 (13)	4 (50)	60	0.0352 ^&,^*
Non-smokers	112 (25)	1 (12)	113	
Unknown	28 (62)	3 (38)	31	
**Differentiation**				
Poor	116 (59)	4 (50)	120	0.6952 ^#^
Moderate	73 (37)	4 (50)	77	
Well	7 (4)	0 (0)	7	
**Histology type**				
SQC	96 (49)	3 (37)	99	0.3936 ^&^
ADC	100 (51)	5 (63)	105	

^&^ Fisher’s Exact test; ^#^ Chi Square test; *: *p* ≤ 0.05. (−): absence of HPV; (+): presence of HPV.

**Table 3 ijms-25-13248-t003:** SLC7A11 gene expression and HPV presence in sub-cohort of 32 lung carcinomas from Chilean patients.

		HPV Presence		
	(−) Cases (%)	(+) Cases (%)	Total	*p*-Value
**TOTAL**	24	8	32	
**SLC7A11 expression**				
Positive	2 (8)	6 (75)	8	0.0080 ^&^
Negative	22 (92)	2 (25)	24	

^&^ Fisher’s Exact test; *p* ≤ 0.05. (−): absence of HPV; (+): presence of HPV.

**Table 4 ijms-25-13248-t004:** SLC7A11 gene expression and smoking in sub-cohort of 32 lung carcinomas from Chilean patients.

		SLC7A11 Gene Expression		
	(−) Cases (%)	(+) Cases (%)	Total	*p*-Value
**TOTAL**	24	8	32	
**Smoking habit**				
Smokers	8 (33)	6 (75)	14	0.0498 ^&^
Non-smokers	16 (67)	2 (25)	18	

^&^ Fisher’s Exact test; *p* ≤ 0.05. (−): absence of HPV; (+): presence of HPV.

## Data Availability

The data of this study will be available upon request.

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
