# Peer review of "SLC7A11 Expression Is Up-Regulated in HPV- and Tobacco-Associated Lung Cancer"

_ijms, 2024, doi:10.3390/ijms252413248_

Round 1

Reviewer 1 Report (Previous Reviewer 1)

Comments and Suggestions for Authors

Overall the manuscript is well written. It provides further proof of the overexpression of SLC7A11 in lung cancer smokers and HP-positive cases. However the study has some major flaws:

1)      There is a high discrepancy between HPV-positive samples (n=8) and HPV-negative (n=24). This does not allow to observe all the heterogeneity of SLC7A11 expression in HPV-positive samples. More HPV-positive samples should be included or the authors should acknowledge this limitation of the study.

2)      The in vitro studies are insufficient. At least two cell types should be used. One normal cell line (BEAS-2B) was already included, but a second lung cancer cell line should be also analyzed.  

3)      Why was the culture media of BEAs-2B changed? The usual recommendation is to use Airway Epithelial Cell Basal Medium and and Bronchial Epithelial Cell Growth Kit or BEGM™ Bronchial Epithelial Cell Growth Medium BulletKit™. I am especially concern with the amount of FBS that was used and that could change the phenotype and molecular characteristics of the normal cell line , as concluded by Malm S.W. el. al. ( DOI: 10.1016/j.tiv.2018.04.008)

Minor observations:

-          Line 42 – please give more concrete information about the HPV found in lung carcinomas. The range 0-100% is too wide.

-          Table 2- please include the explanation that (-) stands for the absence of HPV and (+) stands for the presence of HPV.  

-          Table 3 – include explanations of (-) and (+), the same in Table 4

-          Figure 2: Include the name of the protein above the picture

-          2.6 survival rate- it would be interesting to analyze separately HPV-SLC7A11+ versus HPV-SLC7A11- and HPV+SLC7A11+ versus HPV+SLC7A11-. Also did you also perform a survival analysis based on smoking status? Please mention those results as well.

Author Response

Overall, the manuscript is well written. It provides further proof of the overexpression of SLC7A11 in lung cancer smokers and HP-positive cases. However, the study has some major flaws. There is a high discrepancy between HPV-positive samples (n=8) and HPV-negative (n=24). This does not allow to observe all the heterogeneity of SLC7A11 expression in HPV-positive samples. More HPV-positive samples should be included, or the authors should acknowledge this limitation of the study

Answer: We appreciate your observation highlighting the significant disparity between HPV-positive (n=8) and HPV-negative (n=24) samples. This discrepancy is likely due to the low prevalence of HPV-positive lung cancer cases, limiting our ability to fully characterize the heterogeneity of SLC7A11 expression in this subgroup. While additional HPV-positive samples would be beneficial, resource constraints, including financial limitations, time requirements, and ethical considerations, preclude further analysis of new lung cancer specimens at this time. We added a new paragraph acknowledging our limitations.

The in vitro studies are insufficient. At least two cell types should be used. One normal cell line (BEAS-2B) was already included, but a second lung cancer cell line should be also analyzed.  

Answer: Many thanks for this observation. We added data from a second cell line (A549).

3)      Why was the culture media of BEAs-2B changed? The usual recommendation is to use Airway Epithelial Cell Basal Medium and and Bronchial Epithelial Cell Growth Kit or BEGM™ Bronchial Epithelial Cell Growth Medium BulletKit™. I am especially concern with the amount of FBS that was used and that could change the phenotype and molecular characteristics of the normal cell line, as concluded by Malm S.W. el. al. ( DOI: 10.1016/j.tiv.2018.04.008)

Answer: Many thanks for this observation. DMEM with 10% FBS has been widely used in cell culture for decades, including for BEAS-2B cells. Numerous studies have successfully used this media to analyze cell proliferation and growth. This approach offers a well-established and reproducible method. BEAS-2B cells are relatively robust and can adapt to various culture conditions. While specialized media may provide optimal growth conditions, DMEM with 10% FBS can still support robust proliferation and growth, especially when the focus is on these specific parameters

Minor observations:

-          Line 42 – please give more concrete information about the HPV found in lung carcinomas. The range 0-100% is too wide.

Answer: This was done.

-          Table 2- please include the explanation that (-) stands for the absence of HPV and (+) stands for the presence of HPV.  

Answer: This was done.

-          Table 3 – include explanations of (-) and (+), the same in Table 4

Answer: This was done.

-          Figure 2: Include the name of the protein above the picture

Answer: This was done.

-          2.6 survival rate- it would be interesting to analyze separately HPV-SLC7A11+ versus HPV-SLC7A11- and HPV+SLC7A11+ versus HPV+SLC7A11-. Also did you also perform a survival analysis based on smoking status? Please mention those results as well.

Answer: This data was included.

Reviewer 2 Report (New Reviewer)

Comments and Suggestions for Authors

This work found that the SLC7A11 overexpression was correlated with the HPV- and tobacco-associated lung carcinomas based on the specimen from 204 patients. Although only 8/204 patients have been confirmed with the SLC7A11 overexpression, this phenomenon and potential pathological mechanism is also deserving the exploitation. After my careful reading, many points deserve to be noted for the easier readability, especially in the aspect of FIGURE QUALITY.

1. The authors pointed out that C3, EGF, SERP1, SLC7A11 and SYPL1 genes were up-regulated. Why just choose SLC7A11 for the analysis? The reason should be provided. Also, the citations should be added when previous results are presented in line 85-87.

2. According to Table 3 and 4, the percentage of HPV+/SLC7A11+ and Smoker+/SLC7A11+ is the same. How is the percentage of Smoker+/HPV+/SLC7A11+? Some details should be described more clearly.

3. All the Figures should be improved with higher resolution.

4. In Figure 2, why only select four specimens for detection? The “pb” should be corrected as “bp”.

5. The expression fold change from Figure 3 and 5 should be described in the text. Like 30 x

6. Is SLC7A11 possible viewed as a anticancer target for drug development against Smoker+/HPV+/SLC7A11+ lung carcinomas? It can add some discussions to stimulate more thoughts for readers.

Author Response

Comments and Suggestions for Authors

This work found that the SLC7A11 overexpression was correlated with the HPV- and tobacco-associated lung carcinomas based on the specimen from 204 patients. Although only 8/204 patients have been confirmed with the SLC7A11 overexpression, this phenomenon and potential pathological mechanism is also deserving the exploitation. After my careful reading, many points deserve to be noted for the easier readability, especially in the aspect of FIGURE QUALITY.

  1. The authors pointed out that C3, EGF, SERP1, SLC7A11 and SYPL1 genes were up-regulated. Why just choose SLC7A11 for the analysis? The reason should be provided. Also, the citations should be added when previous results are presented in line 85-87.

Answer: This was done

  1. According to Table 3 and 4, the percentage of HPV+/SLC7A11+ and Smoker+/SLC7A11+ is the same. How is the percentage of Smoker+/HPV+/SLC7A11+? Some details should be described more clearly.

Answer: This was done

  1. All the Figures should be improved with higher resolution.

ANSWER: The resolution of figures was improved.

  1. In Figure 2, why only select four specimens for detection? The “pb” should be corrected as “bp”.

ANSWER: This was corrected

  1. The expression fold change from Figure 3 and 5 should be described in the text. Like 30 x

ANSWER: This was corrected

  1. Is SLC7A11 possible viewed as a anticancer target for drug development against Smoker+/HPV+/SLC7A11+ lung carcinomas? It can add some discussions to stimulate more thoughts for readers.

ANSWER: This was corrected

Round 2

Reviewer 2 Report (New Reviewer)

Comments and Suggestions for Authors

Well solved for publication in IJMS. The "pb" in the DNA marker of Figure 2B should be corrected as "bp".

Author Response

Well solved for publication in IJMS. The "pb" in the DNA marker of Figure 2B should be corrected as "bp".

ANSWER: Many thanks for this observation. This correction was done.

This manuscript is a resubmission of an earlier submission. The following is a list of the peer review reports and author responses from that submission.

Round 1

Reviewer 1 Report

Comments and Suggestions for Authors

The manuscript has an interesting insight into the potential stimulated expression of SLC7A11 in specific cases of lung cancer. However, this is not supported by the data provided. The study has a number of major flaws that the authors need to consider:

-          There not enough data, almost all the results come from the use of online databases and even in this case there is no statistical significance in many cases

-          Generally, the study needs to be more focused, either smoking status and SLC7A11 or HPV infection and SLC7A11

-          I do not see the point in presenting the expression of SLC7A11 in all cancers from TCGA ( 2.1 section from results) – just present the ones that refer to lung cancer

-          The statement from 2.2 – “2.2. SLC7A11 transcripts are up-regulated in both HPV-positive and tobacco smoke-associated lung carcinomas from the TCGA database” is false, there was no statistical significance between HPV-positive and HPV-negative cases

-          The meta-analysis should be mentioned as a supplementary, it is not so relevant as to present it in the study

-          Results 2.3 – the sample size is by far too low for HPV positive cases. You can not draw a conclusion by comparing one group with 4 cases and the other with 24 cases

-          The results from 2.5 section are interesting and the authors should continue on this trajectory. The HPV transfected cells should be compared with the wild type BEAS-2B and see the viability, invasion/migration and colony formation potential. The SLC7A11 expression should also be experimentally manipulated and further tested to observe the exact role of this protein in lung cancer. Also. BEAS-2B is a normal lung cell line, not a lung cancer cell line. A lung cancer cell line transfected with pLXSNE6/E7 vector and compared with the wild type would also be more relevant for the present study.

In conclusion, I recommend that the authors focus solely on the HPV positive versus negative cases, include more tumor in the analysis and do more laboratory tests before this study could be consider for publication

Comments on the Quality of English Language

The English language was fine. Just minor corrections are needed. 

Reviewer 2 Report

Comments and Suggestions for Authors

Human papillomavirus (HPV) is confirmed to be an important pathogenic factor of certain benign and malignant lesions in humans.  However, to date, no definitive conclusions have been drawn on whether HPV infection is associated with lung cancers.

In this article, the authors propose to use SLC7A11 as a potential biomarker for both HR-HPV- and tobacco smoke-associated lung cancers. They first compared SLC7A11 gene expression between cancer samples and paired normal tissues in a publicly available database. Then tried to determine the relationship between SLCA11 expression and HPV presence or smoking history. In the end, they found that overexpression of HPV E6/E7 genes could upregulate SLC7A11 gene expression.

The organization of this article is clear and the statistical analysis method used in different assays is appropriate. However, the writing of the main text should be carefully reviewed to avoid typos as in the abstract “worlwwide”. In the very brief introduction section, the authors failed to raise scientific questions regarding the reason for beginning their research. The results obtained in 2.3 and 2.4 returned no significance in determining the association between SLC7A11 expression and HPV or smoking and thus didn’t support the conclusion drawn in the abstract. This study failed to provide more insight into lung cancer biomarker investigation.  

1. In Result 2.1:

“SLC7A11 expression was significantly increased in lung carcinomas when compared to non-tumor tissues”

SLC7A11 is already reported to be overexpressed in multiple cancer types, including lung cancer, TNBC, PDAC, renal cell carcinoma, liver cancer, and glioma, and its high expression often correlates with poor prognosis.1 Research with a similar method and conclusion has been reported2

“oral and ovary cancers showed a lower SLC7A11 expression than in normal tissues.”

No data support in the main text.

“Additionally, we compared SLC7A11 levels among different histological types of lung cancer, including AdCs, LCCs and SCCs.”

I didn’t see the point in comparing SLC7A11 expression levels among different lung cancer subtypes. Instead, it would make more sense to compare each subtype of lung cancer with normal tissue to reveal their connection with SLC7A11.

2. In Result 2.2:

Figure 3 left panel: case number difference between two groups of samples is too huge which may influence the power of statistical analysis

Figure 3 right panel: The increase of SLC7A11 in the smoking group is marked with “*” and noted P<0.05 in the figure legend. But the main text described this as “not statistically significant”?

The authors conducted another meta-analysis in this section trying to draw the same conclusion as was already stated in section 2.1. Also, based on my trial on the same website, only squamous cell carcinoma shows statistical significance between tumor and normal tissues.

P=0.05197 didn’t support the hypothesis that SLCA11 expression is associated with the patient HPV infection history.

3. In Result 2.3:

Table 1: HPV- patient number =24, HPV+ patient number = 4, and the total patient number = 26 ?

P=0.05197 didn’t support the hypothesis that SLCA11 expression is associated with the patients’ smoking history.

The patient sample source is vague.

4. In Result 2.4:

The authors stated that they used pLXSNHPV16E6/E7 to over-express HPV E6 and E7 genes. However, in the Material and Method section, another vector MSCVBARF1 expressing EBV BARF1 gene was used for cell transfection.

1.               Koppula, P., Zhuang, L. & Gan, B. Cystine transporter SLC7A11/xCT in cancer: ferroptosis, nutrient dependency, and cancer therapy. Protein & Cell 12, 599–620 (2021).

2.               Ji, X. et al. xCT (SLC7A11)-mediated metabolic reprogramming promotes non-small cell lung cancer progression. Oncogene 37, 5007–5019 (2018).